# Action-Quantized Offline Reinforcement Learning for Robotic Skill Learning

**Jianlan Luo   Perry Dong   Jeffrey Wu   Aviral Kumar   Xinyang Geng   Sergey Levine**
University of California, Berkeley

**Abstract:** The offline reinforcement learning (RL) paradigm provides a general recipe to convert static behavior datasets into policies that can perform better than the policy that collected the data. While policy constraints, conservatism, and other methods for mitigating distributional shifts have made offline reinforcement learning more effective, the continuous action setting often necessitates various approximations for applying these techniques. Many of these challenges are greatly alleviated in discrete action settings, where offline RL constraints and regularizers can often be computed more precisely or even exactly. In this paper, we propose an adaptive scheme for action quantization. We use a VQ-VAE to learn state-conditioned action quantization, avoiding the exponential blowup that comes with naïve discretization of the action space. We show that several state-of-the-art offline RL methods such as IQL, CQL, and BRAC improve in performance on benchmarks when combined with our proposed discretization scheme. We further validate our approach on a set of challenging long-horizon complex robotic manipulation tasks in the Robomimic environment, where our discretized offline RL algorithms are able to improve upon their continuous counterparts by 2-3x. Our project page is at `saqrl.github.io`

**Keywords:** Offline Reinforcement Learning, Discretization

## 1   Introduction

Offline reinforcement learning (RL) aims to learn policies from static databases of previously collected experience. As opposed to pure imitation, offline RL holds the promise to transform logged datasets into policies that perform *better* than the behavior policy that collected the dataset by maximizing task rewards. The key challenge in offline RL is the overestimation of the value of actions that were not seen in the dataset, which destabilizes training and leads to policies that perform much worse than their value function estimates would suggest. To address this issue, a wide variety of methods have been proposed recently [1, 2, 3, 4, 5]. Typically, these methods employ some mechanism to stay "close" to behavior policies, such as policy constraints or value conservatism.

While these methods in principle address overestimation and distributional shift, in practice implementing these approaches requires various approximations (depending on the method) that can lead to hyperparameter sensitivity or performance that is worse than their theoretical formulation might suggest. This issue is particularly pronounced on "narrow" datasets that consist of relatively deterministic behavior, such as the demonstration data that is often used in robotic learning. Our key observation is that these issues can be mitigated by properly discretizing continuous action spaces, and then employing discrete action versions of these methods, where many of these approximations are unnecessary. For example, policy constraints or any conservatism regularizer that requires expectations over actions can be computed exactly with discrete actions.

Unfortunately, naïvely discretizing the action space can result in an exponential blowup in the number of actions, while coarse adaptive discretization methods can lead to imprecise actions. In this paper, we propose an adaptive scheme, state-conditioned action quantization (SAQ), for discretizaing

7th Conference on Robot Learning (CoRL 2023), Atlanta, USA.

continuous action spaces. We perform state-conditioned action discretization by utilizing a VQ-VAE model. SAQ is based on a simple insight: given a particular dataset, there are often only a few in-distribution options available in each state, corresponding roughly to the "primitives" that are supported under the data. This allows us to use comparatively very small discrete action spaces, without suffering from the curse of dimensionality, while still enjoying the benefits of discrete action spaces and simpler offline RL algorithm implementations.

The main contribution of this work is a practical approach, SAQ, for learning quantized action representations for improving continuous-action offline RL methods on a variety of robotic learning tasks. We present a general method for learning state-conditioned action discretizations and then apply this method with three offline RL methods: conservative Q-learning (CQL) [3], implicit Q-learning (IQL) [4], and behavior regularized actor-critic (BRAC) [1]. All of these methods require some sort of approximation with continuous actions, while the discrete version provides for a convenient implementation that avoids such approximations. We find that the discrete version of each method implemented with SAQ generally yields improved performance on commonly used benchmark tasks over each method's continuous-action counterpart, particularly on "narrow" datasets (e.g., expert data). We also evaluate these methods on a set of challenging robotic manipulation tasks from the Robomimic environment [6], where continuous offline RL methods struggled to get good performance, and find that our approach outperforms both prior offline RL methods in this setting by a large margin.

## 2  Related Work

**Offline RL.** Some offline RL methods use a policy constraint to mitigate overestimation from distributional shift, constraining the learned policy to stay close to the data via density modeling or some other divergence measure [1, 2, 7, 8, 9, 10, 11, 12, 13, 4, 5, 14]. Another line of work directly regularizes Q values of unseen actions, which we refer as conservative value function methods [3, 15, 16, 17, 18, 19, 20]. While both types of methods can be formulated in ways that in principle address distributional shift, in practice they require some kind of approximation to be applied with continuous actions, either in estimating the behavior policy, or in computing integrals or expectations over the action space to formulate pessimism penalties. This introduces approximation errors and complexity. We show that our approach can avoid the need for such approximations for three representative methods from both categories, leading to improved performance.

**Discretizing continuous action space for control.** Recent work has shown discretizing continuous actions can yield good performance, and several discretization strategies have been introduced. To address the issue of exponential growth of actions in the naïve discretization scheme, one line of methods assumes independence of action dimensions [21, 22, 23, 24, 25, 26], or perform discretization in an autoregressive way [27, 28, 29]. Our work is different in that we perform state-dependant discretization, which adaptively controls the precision of the discretization scheme. Perhaps the closest to our work is Dadashi et al. [30], which also learns a state-dependant discretization. It assumes access to a human demonstration dataset. Given the current state, it uses a neural network to index a set of plausible continuous actions and trains the network by comparing its output with the demonstration data. Our work differs in that we focus on the offline RL setting, where discretization could enforce conservatism and policy constraints exactly. To our knowledge, our work is the first to propose adaptive action discretization with VQ-VAEs for offline RL.

## 3  Preliminaries

The RL problem is formally defined by a Markov decision processes (MDPs) $\mathcal{M} = (\mathcal{S}, \mathcal{A}, T, r, \mu_0, \gamma)$, where $\mathcal{S}$ and $\mathcal{A}$ denote the state and action spaces, and $T(\mathbf{s}'|\mathbf{s}, \mathbf{a})$, $r(\mathbf{s}, \mathbf{a})$ represent the dynamics and reward function, respectively. $\mu_0(\mathbf{s})$ denotes the initial state distribution, and $\gamma \in (0, 1)$ denotes the discount factor. The objective of RL is to learn a policy that maximizes the return (discounted sum of rewards): $\max_\pi J(\pi) := \mathbb{E}_{(\mathbf{s}_t, \mathbf{a}_t) \sim \pi}[\sum_t \gamma^t r(\mathbf{s}_t, \mathbf{a}_t)]$. In offline RL, we are provided with an offline

dataset $\mathcal{D} = \{(\mathbf{s}, \mathbf{a}, r, \mathbf{s}')\}$ of transitions collected using a behavior policy $\pi_\beta$, and our goal is to find the best possible policy only using the given dataset, without any additional online data collection. In this paper, we focus on three offline RL methods, conservative Q-learning (CQL), implicit Q-learning (IQL), and behavior regularized actor-critic(BRAC); though our approach could likely be extended to other methods also.

**Conservative Q-learning.** Naïvely learning a $Q$-value function from the offline dataset (e.g., via Q-learning or FQI) suffers from OOD actions [31, 2, 32], and the CQL algorithm [3] applies a regularizer $\mathcal{R}(\theta)$ to prevent querying the target Q-function on unseen actions. $\mathcal{R}(\theta)$ minimizes the Q-values under the policy $\pi(\mathbf{a}|\mathbf{s})$, and counterbalances this term by maximizing the values of the actions in $\mathcal{D}$. Formally, we have the following objective during Bellman backup:

$$\min_\theta \frac{1}{2} \mathbb{E}_{\substack{\mathbf{s}, \mathbf{a}, \mathbf{s}' \sim \mathcal{D} \\ \mathbf{a}' \sim \pi}} \left[ \left( Q_\theta(\mathbf{s}, \mathbf{a}) - r - \gamma \bar{Q}(\mathbf{s}', \mathbf{a}') \right)^2 \right] + \alpha \left( \mathbb{E}_{\mathbf{s} \sim \mathcal{D}, \mathbf{a} \sim \pi} [Q_\theta(\mathbf{s}, \mathbf{a})] - \mathbb{E}_{\mathbf{s}, \mathbf{a} \sim \mathcal{D}} [Q_\theta(\mathbf{s}, \mathbf{a})] \right), \quad (1)$$

where $\bar{Q}$ denotes the target $Q$-function.

**Behavior regularized actor-critic.** The specific version BRAC-v explicitly subtracts the divergence $D(\pi(\cdot|\mathbf{s}'), \pi_\beta(\cdot|\mathbf{s}'))$ from the target value while performing the Bellman update. Additionally, since the divergence between the learned policy and the behavior policy *at the current state* is not a part of the Q-function, BRAC-v also explicitly adds the divergence value at the current state to the policy update. We instantiate the version of BRAC that uses the KL-divergence:

$$D_{\mathrm{KL}}(\pi(\cdot|\mathbf{s}), \pi_\beta(\cdot|\mathbf{s})) = \mathbb{E}_{\mathbf{a} \sim \pi(\cdot|\mathbf{s})} [\log \pi(\mathbf{a}|\mathbf{s}) - \log \pi_\beta(\mathbf{a}|\mathbf{s})]. \quad (2)$$

To estimate the second term $\log \pi_\beta(\mathbf{a}|\mathbf{s})$, BRAC trains an additional behavior policy, that we denote as $\hat{\pi}_\beta$. Denoting the policy and the Q-function as $\pi_\phi$ and $Q_\theta$, the BRAC-v perform following Bellman update:

$$\min_\theta \ \mathbb{E}_{\mathbf{s}, \mathbf{a} \sim \mathcal{D}} \left[ \left( r(\mathbf{s}, \mathbf{a}) + \gamma \mathbb{E}_{\mathbf{a}' \sim \pi_\phi(\cdot|\mathbf{s}')} [\bar{Q}_\theta(\mathbf{s}', \mathbf{a}') + \beta \log \hat{\pi}_\beta(\mathbf{a}'|\mathbf{s}')] - Q_\theta(\mathbf{s}, \mathbf{a}) \right)^2 \right], \quad (3)$$

then extracts a policy by:

$$\max_\phi \ \mathbb{E}_{\substack{\mathbf{s} \sim \mathcal{D} \\ \mathbf{a} \sim \pi_\phi(\cdot|\mathbf{s})}} [Q_\theta(\mathbf{s}, \mathbf{a}) + \beta \log \hat{\pi}_\beta(\mathbf{a}|\mathbf{s}) - \alpha \log \pi_\phi(\mathbf{a}|\mathbf{s})] \quad (4)$$

**Implicit Q-Learning.** Instead of applying policy constraints or critic regularizations, IQL uses expectile regression to learn a value function to approximate an expectile $\tau$ over the distribution of actions by optimizing the value objective

$$L_V(\psi) = \mathbb{E}_{(\mathbf{s}, \mathbf{a}) \sim \mathcal{D}} [L_2^\tau (Q_{\hat{\theta}}(\mathbf{s}, \mathbf{a}) - V_\psi(\mathbf{s}))],$$

where $Q_{\hat{\theta}}(\mathbf{s}, \mathbf{a})$ is a parameterized target critic and $L_2^\tau(u) = |\tau - \mathbb{I}(u < 0)|u^2$. This value function is then used to update the Q-function with TD error following the objective function

$$L_Q(\theta) = \mathbb{E}_{(\mathbf{s}, \mathbf{a}, \mathbf{s}') \sim \mathcal{D}} [(r(\mathbf{s}, \mathbf{a}) + \gamma V_\psi(\mathbf{s}') - Q_\theta(\mathbf{s}, \mathbf{a}))^2]$$

With the value function $V_\psi$ and Q-function $Q_{\hat{\theta}}(\mathbf{s}, \mathbf{a})$, the policy is learned using advantage weighted regression following the loss below:

$$L_\pi(\phi) = E_{(\mathbf{s}, \mathbf{a}) \sim D} [\exp(\alpha(Q_{\hat{\theta}}(\mathbf{s}, \mathbf{a}) - V_\psi(\mathbf{s}))) \log \pi_\phi(\mathbf{a}|\mathbf{s})].$$

**Vector quantized variational autoencoders.** Our method uses the VQ-VAE [33] as part of the discretization process. The VQ-VAE consists of an encoder and a decoder. The encoder network parameterizes a posterior distribution $q_\phi(z|x)$ over discrete latent variables $z$ given the input data $x$. The decoder that parameterizes the distribution $p_\theta(x|z)$ then reconstructs the observations from these discrete variables. Specifically, the encoder first outputs a continuous embedding $e_\phi(x) \in R^D$. Discretization is done with a codebook defined on $R^{K \times D}$, composed of $K$ vectors $e_j \in R^D$, where $j = 1, 2, 3...K$. The embedding $e_\phi(x)$ is compared with all the vectors in the codebook, and the

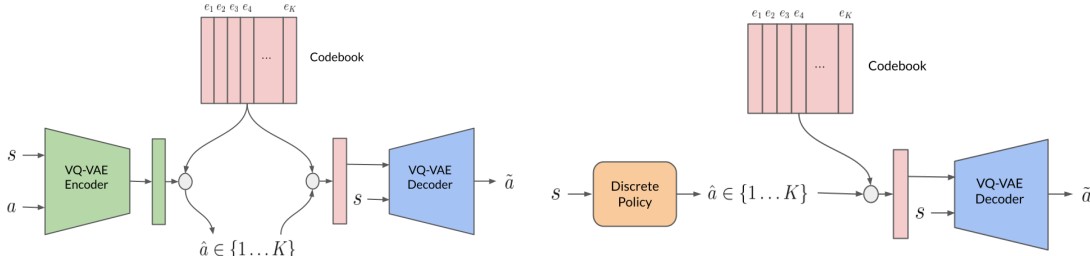

Figure 1: **Left**: Training SAQ. **Right**: Discrete policy training/evaluation with SAQ. SAQ learns a scalar discrete representation of continuous actions using state-conditioned VQ-VAE. During policy training and evaluation time, we use the decoder of SAQ to reconstruct the continuous actions from the discrete policy actions.

discrete state $z$ is chosen to correspond to the nearest codebook entry. The discrete posterior is then defined as:

$$q(z = k|x) = \begin{cases} 1, & \text{if } k = \arg\min_j \|e(x) - e_j\|_2 \\ 0, & \text{otherwise} \end{cases} \tag{5}$$

Let's denote $z_{q_\phi}(x)$ as the final latent code from the encoder, i.e., $z_{q_\phi}(x) = e_k$. The decoder will reconstruct the original input $x$ based on $z_{q_\phi}(x)$. Additionally, a codebook loss and a commitment loss are added to move codebook vectors and encoder embeddings close to each other. The overall training objective is as follows:

$$L(\theta, \phi, x) = \mathbb{E}_{e \in e_j, j=1,2,3...K} \left[ -\log p_\theta(x|z_{q_\phi}(x)) + \|\text{sg}[e_\phi(x)] - e\|_2^2 + \|e_\phi(x) - \text{sg}[e]\|_2^2 \right], \tag{6}$$

where sg stands for the stop gradient operator. We can then optimize the parameters of the encoder and decoder with this objective.

## 4 State Conditioned Action Quantization for Offline RL

The central premise of this paper is that carefully and efficiently discretizing the action space of a continuous-action RL problem can make it significantly simpler to implement offline RL methods based on critic or actor regularization, and that such discretized methods can attain better performance on especially difficult data distributions, such as narrow datasets. How should we discretize a continuous action space? Discretizing too coarsely can make it difficult to perform delicate tasks that require fine control, and can collapse in-distribution and out-of-distribution actions into the same bin, making offline RL more difficult. On the other hand, naïvely using a very fine discretization can be intractable due to the curse of dimensionality. Our key observation is that we can construct a discretization with relatively few discrete actions but with minimal loss of resolution if we employ a learned state-conditioned discretization, which we can accomplish by means of a VQ-VAE model. In this section, we will describe this approach, and discuss how it can be combined with simple and effective discrete-action offline RL methods.

### 4.1 State-Conditioned Action Discretization via Scalar-Quantized Auto-Encoders

Abstractly, we aim to learn a state-conditioned action discretization (SAQ) scheme that can map a continuous action $\mathbf{a}$ at a given state $\mathbf{s}$ to a discrete variable $\widehat{a}$ for training, and then given a choice of the discrete variable $\widehat{a}$ also map it back to the original action space $\mathcal{A}$ for evaluation.

Given this requirement, a natural choice is to utilize auto-encoders that can produce state-conditioned discrete codes for a given action. Therefore, our approach adapts the vector-quantized variational auto-encoder (VQ-VAEs) [33] framework for performing action discretization, with the difference that unlike conventional VQ-VAEs that typically learn a multi-dimensional discrete latent code for an input, we only aim to learn a scalar discrete latent code for a given input action as this is sufficient for performing downstream RL.

More formally, denoting the parameters of the VQ-VAE encoder with $\phi$ and the decoder with $\theta$, we wish to learn an encoder $q_\phi(\cdot|s,a)$ that produces a one-dimensional discrete latent action, and a decoder to map the discrete action code back to the original action space, $p_\theta(a|s,\widehat{a})$. Following the training procedure of VQ-VAEs, we first obtain a latent embedding $z_{q_\phi}(s,a)$ from the approximate posterior $q_\phi(\cdot|s,a)$ and then compare it within the codebook, and obtain the nearest vector $e_k$. Next, we run this latent action code through the decoder to obtain an approximate reconstructed action, $\tilde{a} \sim p(\cdot|s,e_k)$, which is further trained to minimize the reconstruction error against the original action $a$ passed as input to the encoder. Following the training procedure proposed by Van den Oord et al. [33], the overall training objective for SAQ is shown in Equation 7, where sg is the operator of stopping gradient, $\mathcal{D}$ is the offline dataset:

$$\min_{\substack{\theta,\phi \\ e \in e_j, j=1,2,3\dots K}} \mathbb{E}_{\mathbf{s},\mathbf{a}\sim\mathcal{D}} \left[ -\log p_\theta(\mathbf{a}|z_{q_\phi}(\mathbf{s},\mathbf{a})) + \|\mathrm{sg}[e_\phi(\mathbf{s},\mathbf{a})] - e\|_2^2 + \|e_\phi(\mathbf{s},\mathbf{a}) - \mathrm{sg}[e]\|_2^2 \right] \quad (7)$$

## 4.2 Offline RL with Discretized Actions

After training VQ-VAE to obtain a discrete action representation, we can transform the original problem with continuous action into a discrete problem by learning policies and value functions in the discrete action space.

**SAQ-CQL.** For CQL, we follow the discrete variant of CQL [3], where we learn the Q-function with a conservatism penalty and parameterize the policy as $\pi(\mathbf{a}|\mathbf{s}) \propto Q(\mathbf{s},\mathbf{a})$. Specifically, we employ the maximum entropy variant of CQL and replace the sampled log-integral-exp with log-sum-exp by summing over all discrete actions. This allows us to compute the CQL conservatism loss *exactly*, without having to rely on the samples from policy to estimate the integral of Q-function. Our overall objective of SAQ-CQL is therefore

$$\min_\theta \frac{1}{2} \mathbb{E}_{\substack{\mathbf{s},\mathbf{a},\mathbf{s}'\sim\mathcal{D} \\ \mathbf{a}'\sim\pi}} \left[ \left( Q_\theta(\mathbf{s},\mathbf{a}) - r - \gamma\bar{Q}(\mathbf{s}',\mathbf{a}') \right)^2 \right] + \alpha \mathbb{E}_{\mathbf{s},\mathbf{a}\sim\mathcal{D}} \left[ \log\sum_i \exp(Q_\theta(\mathbf{s},\mathbf{a}_i)) - Q_\theta(\mathbf{s},\mathbf{a}) \right].$$

We note that in this specific variant of discrete CQL, the conservative penalty term $\mathbb{E}_{\mathbf{s},\mathbf{a}\sim\mathcal{D}}[\log\sum_i \exp(Q_\theta(\mathbf{s},\widehat{\mathbf{a}}_i)) - Q_\theta(\mathbf{s},\widehat{\mathbf{a}})]$ is exactly equivalent to the discrete negative log-likelihood behavioral cloning loss:

$$\mathbb{E}_{\mathbf{s},\mathbf{a}\sim\mathcal{D}} \left[ \log\sum_i \exp(Q_\theta(\mathbf{s},\widehat{\mathbf{a}}_i)) - Q_\theta(\mathbf{s},\widehat{\mathbf{a}}) \right] = - \mathbb{E}_{\mathbf{s},\mathbf{a}\sim\mathcal{D}} \left[ \log \frac{\exp(Q_\theta(\mathbf{s},\widehat{\mathbf{a}}))}{\sum_i \exp(Q_\theta(\mathbf{s},\widehat{\mathbf{a}}_i))} \right]$$
$$= - \mathbb{E}_{\mathbf{s},\mathbf{a}\sim\mathcal{D}} \left[ \log \pi_\theta(\widehat{\mathbf{a}}|\mathbf{s}) \right]. \quad (8)$$

**SAQ-IQL.** For IQL, we follow the original problem formulation in Kostrikov et al. [4], then derive its closed-form solution in the discrete case. We denote the advantage as $A^\pi(\mathbf{s},\widehat{\mathbf{a}}) = Q^\pi(\mathbf{s},\widehat{\mathbf{a}}) - V^\pi(\mathbf{s})$ for a given policy $\pi$, which can be obtained by the same procedure detailed in Kostrikov et al. [4]. The policy extraction step in IQL objective is to find a policy $\pi^\star$ so that $A^\pi(\mathbf{s},\mathbf{a})$ can be maximized while staying close to behavior policy $\pi_\beta$:

$$\pi^\star = \arg\max \mathbb{E}_{\widehat{\mathbf{a}}\sim\pi(\cdot|\mathbf{s})}[A^\pi(\mathbf{s},\widehat{\mathbf{a}})] \quad \text{s.t.} \quad D_{\mathrm{KL}}(\pi(\cdot|\mathbf{s}),\pi_\beta(\cdot|\mathbf{s})) \leq \epsilon. \quad (9)$$

We can solve this constrained optimization in Eq. 9 by using the Lagrangian method, and the solution is given by:

$$\pi^\star(\widehat{\mathbf{a}}|\mathbf{s}) \propto \exp\left[ \left(\frac{1}{\lambda}A^\pi(\mathbf{s},\widehat{\mathbf{a}}) + \log\pi_\beta(\widehat{\mathbf{a}}|\mathbf{s})\right) \right], \quad (10)$$

where $\lambda$ is the Lagrangian multiplier, which controls the amount of constraint deviation. We refer readers to Appendix A for detailed derivation. In practice, we can obtain $\pi_\beta$ by training an additional BC policy on the behavior dataset.

**SAQ-BRAC**   For SAQ-BRAC, we first learn a categorical discrete-action behavior policy $\pi_\beta(\mathbf{a}|\mathbf{s})$ on the discretized actions by maximizing the log-likelihood of the offline dataset. We then implement discrete BRAC by exactly following Eq. 3 and Eq. 4, where we compute the KL-divergence between the learned policy and behavior policy by simply summing over all discrete actions.

### 4.3   Diagnosing offline RL performance with constraint enforcement

As conjectured in Sec. 1, we associate inexact constraint enforcement with the degradation of performance of offline RL methods. To verify this hypothesis, in this section, we conduct an experiment in a pointmass navigation environment (maze2d-large from Fu et al. [34]), where the RL agent controls a point mass to navigate through the maze from a fixed starting point to the goal point, as shown on the left side of Figure 2. We collect three optimal demonstration trajectories as the offline dataset and run both continuous CQL and SAQ-CQL on this domain. We visualize the average return in the middle as well as the sample estimated (shown in blue) and the exact CQL conservatism penalty (shown in green) on the right of Figure 2. We observe that the performance of continuous CQL rapidly degrades with more training steps, with the estimated conservative penalty $\mathcal{R}(\theta)$ diverging from the exact value in the period where the performance degrades. The error in estimation causes the policy training to overly penalize the wrong action samples, resulting in degraded performance. On the contrary, SAQ-CQL computes the penalty exactly, which enables the optimizer to smoothly minimize the penalty, leading to robust policy performance. While it may be that with more hyperparameter tuning the baseline continuous-action version would perform better, the results suggest that poor performance in this case corresponds to the inexact estimate of the CQL regularizer, which is largely mitigated by our discretization approach.

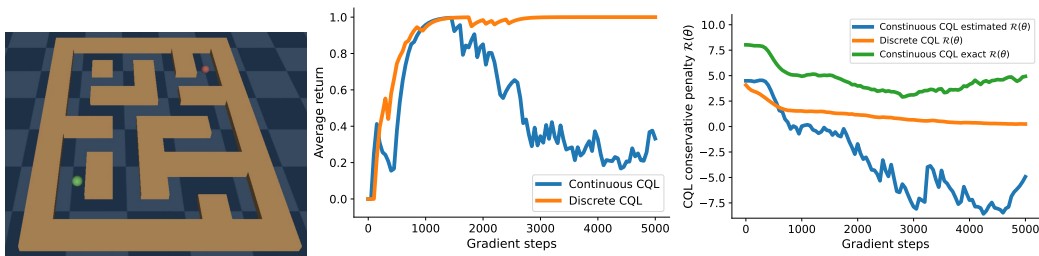

Figure 2: **Left**: visualization of the maze2d environment. **Middle**: the average return of SAQ-CQL and continuous CQL over the course of offline training. **Right**: estimated and exact CQL conservatism penalty values. The estimated conservative penalty for continuous-action CQL diverges significantly from the true value during training, resulting in rapid performance degradation, while the SAQ-CQL enjoys stable training by optimizing the exact penalty value. As a result, the training process (middle) is significantly more stable with SAQ, which we expect should make the algorithm easier to use, simplifying checkpoint selection and tuning.

## 5   Experiments

Our experiments compare three continuous-action offline RL methods to their discretized variants instantiated with SAQ, both on standard offline RL benchmarks and the Robomimic robotic manipulation environment [6]. We combine SAQ with: CQL [3], IQL [5], and BRAC [1]. We first evaluate on the D4RL [35] suite of tasks, and then use the Robomimic tasks [6], which prior work has found to present a particular challenge for offline RL methods. Appendix B presents a detailed analysis and ablation study of SAQ to understand how it affects offline RL training, which we encourage readers to examine for a more detailed study of the method. We also present a comparison with Aquadem [30] in Appendix C.

### 5.1   D4RL Benchmark Evaluations

We present the results on the D4RL benchmark suite [34] in Table 5, with illustrations of the evaluation domains shown in Figure 3. Some of the benchmark tasks consist of narrow data distributions, while

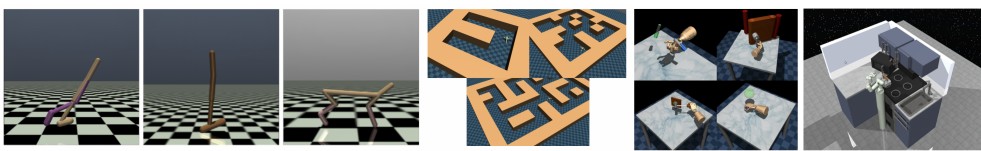

Figure 3: D4RL benchmark tasks: locomotion, antmaze, adroit and kitchen.

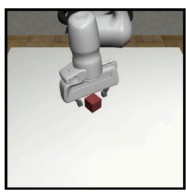 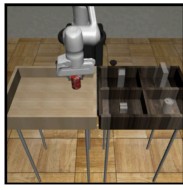 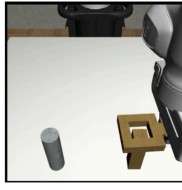 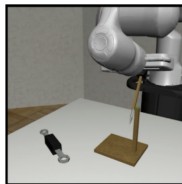 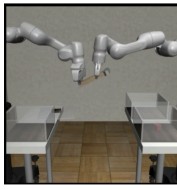

Figure 4: Robomimic tasks: `lift`, `can`, `square`, `tool-hang`, `transport`. Among these tasks, `square` features multi-modal behavior, `tool-hang` requires high-precision manipulation, `transport` is particularly challenging because its long-horizon complex multi-modal nature.

others contain high-coverage data. For example, the *"kitchen-"* domains consist of data from expert human teleoperators controlling a robotic arm to perform a variety of manipulation skills, which must then be sequenced and recombined by the algorithm to solve specific tasks. The *"-expert"* and *"-medium-expert"* datasets for the locomotion tasks also contain narrow expert data, possibly in combination with broad but suboptimal data, with the *"-medium-expert"* version presenting a particular challenge in terms of picking out the narrow but high-performing expert mode from the noisier overall distribution. The *"adroit"* dexterous manipulation tasks also contain narrow expert demonstration data from humans controlling the simulated robotic hand with a data glove. The narrow datasets are particularly challenging for prior continuous-action methods, as the narrow behavior policies exacerbate challenges due to imperfect approximations for policy constraints and conservative regularizers. This is particularly pronounced for high-dimensional domains, such as the 24-DoF *"adroit"* robotic hand, where methods that require integrals or expectations over the action space (such as the regularizer in CQL) must sample and average over a high-dimensional action space. We see that for each algorithm (BRAC, IQL, and CQL), the version of each algorithm discretized with SAQ improves with respect to the average score in each of the domain types (locomotion, antmaze, adroit, and kitchen). Although in some cases the improvement is small, in other cases it is very significant, particularly for narrow-data adroit and kitchen tasks, and the *"-medium-expert"* versions of the halfcheetah and hopper task. We hypothesize that SAQ performs well in these domains because the discretization can capture the individual (narrow) modes, while the discrete-action RL algorithm can then select from among these modes to attain the best performance.

While SAQ can lead to significant improvements over the continuous-action counterpart of each method, the main benefit of SAQ is not in raw performance, but in terms of the simplification of the downstream RL problem. Our experiments provide an "apples to apples" comparison for each RL method, comparing its discrete (SAQ) version to its original continuous formulation, but of course, other offline RL methods might perform better on some tasks, and we do not aim to show that SAQ achieves state-of-the-art performance over all possible offline RL techniques. However, we expect that the simple discrete-action RL problem that is presented via our proposed discretization will in the long run make it easier to scale offline RL to harder and more complex problem domains, and provide a valuable tool to the RL practitioner. RL methods tend to be complex and difficult to tune, so any improvement that simplifies the RL part of the problem is likely to improve practical utility.

## 5.2 Robomimic Evaluation

Robomimic [6] is a set of environments and demonstration datasets that require controlling a 7-DoF robot arm to perform a variety of manipulation tasks. Prior work [6] reported that these environments are particularly challenging for offline RL algorithms, with the best-reported results obtained by

| Task | | BRAC | SAQ-BRAC | IQL | SAQ-IQL | CQL | SAQ-CQL | BC | SAQ-BC |
|---|---|---|---|---|---|---|---|---|---|
| locomotion avg | | 64.16 | **66.98** | 75.14 | **76.67** | 75.6 | **77.35** | 38.86 | **60.77** |
| antmaze avg | | 18.84 | **29.5** | 55.34 | **55.92** | 55.15 | **56.87** | 0 | 0 |
| adroit avg | | 23.86 | **40.09** | 20.32 | **22.82** | 10.58 | **21.37** | 12.35 | **21** |
| kitchen avg | | 15.78 | **36.11** | 50.83 | **58.61** | 48.67 | **65.89** | 17.22 | **57** |

Table 1: Averaged normalized scores across locomotion, Adroit, AntMaze, and kitchen domains from D4RL. The version of each algorithm discretized with SAQ generally improves over the average score of the original algorithm in each class of domains, with particularly pronounced improvements on narrow dataset domains such as adroit and kitchen. The full results are deferred to Appendix C

| Task | | IQL | SAQ-IQL | CQL | Robomimic CQL | SAQ-CQL | BC | Robomimic BC | SAQ-BC |
|---|---|---|---|---|---|---|---|---|---|
| lift | | 58 | 90 | 64.2 | 92.7 | 90.8 | 59.47 | 100 | 90.13 |
| can | | 33.73 | 68 | 19.6 | 38 | 71.2 | 31.73 | 95.3 | 66.4 |
| square | | 26.93 | 46.67 | 0 | 5.3 | 44.27 | 19.33 | 78.7 | 45.33 |
| tool-hang | | 2.67 | 28 | 0 | 0 | 3.87 | 1.87 | 17.3 | 3.47 |
| transport | | 0 | 2 | 0 | 0 | 3.47 | 0.27 | 29.3 | 3.2 |
| average | | 24.27 | **46.93** | 16.76 | 27.2 | **42.72** | 22.53 | **64.12** | 41.71 |

Table 2: Average success rates on Robomimic tasks using the Proficient Human dataset for each task.

simpler imitation learning methods. Using the "proficient human" (PH) datasets, which each consist of 200 successful trajectories with a binary reward, we trained policies with the continuous version of IQL, CQL, and BC, as well as discrete-action policies using the discretization from SAQ. The results, presented in Table 2, show that SAQ indeed improves continuous offline RL by large margins on all the tasks considered. It's worthwhile to mention that our continuous BC results don't exactly match the original paper [6], as the authors point out, they adopt a Gaussian Mixture Model (GMM) for the policy class and extensively optimize parameters then perform checkpoint selection, whereas we directly train unimodal BC policies.

While the D4RL results suggest that discretization with SAQ can consistently improve the performance of each offline RL algorithm, these results further suggest that domains that are especially challenging for offline RL, such as narrow demonstration datasets, are particularly amenable for SAQ, where it can enable offline RL methods that were previously outperformed by simple imitation learning to attain significantly better results. This suggests that SAQ can be a particularly effective tool in robotic learning, where narrow demonstration datasets might be commonplace.

# 6 Discussion, Limitations, and Future Work

We presented a method for state-conditioned action quantization to improve continuous offline RL algorithms. Our approach allows offline RL methods to enforce policy constraints or value conservatism more exactly as compared to their continuous counterparts. This is particularly relevant and important in the robotic learning setting where we usually assume narrow datasets of expert demonstrations where function approximation errors get even more exaggerated. However our approach does have a number of limitations. First, we require sufficient state-action coverage to be able to perform state-conditioned action quantization. This is a common assumption in offline RL and is true in many curated datasets; however, it might be challenging to obtain such datasets in real-world robotic settings. That said, our approach is able to achieve impressive performance in the Robomimic environment which largely composed of real human teleoperation data. Second, it is not clear the best way to adopt our method in the online finetuning setting, where new data might invalidate the learned discretization. Adaptively adjusting the discretization during online training could be a valuable topic to explore in future work.

**Acknowledgments**

This research was partly supported through the Office of Naval Research through N00014-21-1-2838 and N00014-20-1-2383. We acknowledge computing support from the Berkeley Research Computing(BRC) program and the NSF Cloudbank program.

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

# Appendix

## A  Detailed SAQ-IQL derivation

In this section, we describe in detail the derivation for the SAQ-IQL algorithm. We start with the original optimization objective from IQL that employs an explicit policy constraint w.r.t. to a behavior policy

$$\pi^* = \arg\max_{\pi \in \Pi} \mathbb{E}_{\mathbf{a} \sim \pi(\cdot|\mathbf{s})}[A^\pi(\mathbf{s}, \mathbf{a})]$$

$$\text{s.t. } D_{KL}(\pi(\cdot, \mathbf{s})||\pi_\beta(\cdot|\mathbf{s})) \leq \epsilon \tag{11}$$

$$\sum_{\mathbf{a}} \pi(\mathbf{a}|\mathbf{s}) = 1$$

We can write down the Lagrangian of 11 and expand it out:

$$L(\pi, \lambda, \alpha) = \mathbb{E}_{\mathbf{a} \sim \pi(\cdot|\mathbf{s})}[A^\pi(\mathbf{s}, \mathbf{a})] + \lambda(\epsilon - D_{KL}(\pi(\cdot|\mathbf{s})||\pi_\beta(\cdot|\mathbf{s}))) + \alpha(1 - \sum_{\mathbf{a}} \pi(\mathbf{a}|\mathbf{s}))$$

$$L(\pi, \lambda, \alpha) = \sum_{a \sim \pi(\cdot|s)} (\pi(\mathbf{a}|\mathbf{s})A^\pi(\mathbf{s}, \mathbf{a})) + \lambda(\epsilon - \sum_{\mathbf{a} \sim \pi(\cdot|\mathbf{s})} (\pi(\mathbf{a}|\mathbf{s}) \log(\frac{\pi(\mathbf{a}|\mathbf{s})}{\pi_\beta(\mathbf{a}|\mathbf{s})}))) + \alpha(1 - \sum_{\mathbf{a}} \pi(\mathbf{a}|\mathbf{s}))$$

Differentiating $L(\pi, \lambda, \alpha)$ with respect to $\pi(\mathbf{a}|\mathbf{s})$ results in

$$\frac{\partial L(\pi, \lambda, \alpha)}{\partial \pi} = A^\pi(\mathbf{s}, \mathbf{a}) - \lambda(\log(\pi(\mathbf{a}|\mathbf{s})) - \log(\pi_\beta(\mathbf{a}|\mathbf{s})) + 1) - \alpha$$

We divide the differentiated Lagrangian by the constant $\lambda$

$$\frac{\partial L(\pi, \lambda, \alpha)}{\partial \pi} \cdot \frac{1}{\lambda} = \frac{A^\pi(\mathbf{s}, \mathbf{a})}{\lambda} + \log(\pi_\beta(\mathbf{a}|\mathbf{s})) - \log(\pi(\mathbf{a}|\mathbf{s})) - 1 - \frac{\alpha}{\lambda}$$

and set the resulting expression to 0 to arrive at $\pi^*(\mathbf{a}|\mathbf{s})$,

$$\pi^*(\mathbf{a}|\mathbf{s}) = \exp[\frac{A^\pi(\mathbf{s}, \mathbf{a})}{\lambda} + \log(\pi_\beta(\mathbf{a}|\mathbf{s})) - 1 - \frac{\alpha}{\lambda}]$$

$$\pi^*(\mathbf{a}|\mathbf{s}) = \frac{1}{z(\mathbf{s})} \exp[\frac{A^\pi(\mathbf{s}, \mathbf{a})}{\lambda} + \log(\pi_\beta(\mathbf{a}|\mathbf{s}))]$$

where $z(\mathbf{s})$ is a normalizing term.

# B   Ablation Studies for SAQ

| Task | No State SAQ-CQL | State SAQ-CQL | No State SAQ-IQL | State SAQ-IQL | No State SAQ-BRAC | State SAQ-BRAC |
|---|---|---|---|---|---|---|
| halfcheetah-medium-replay-v2 | 1.56 | **47.07** | 1.56 | **36.2** | -1.6 | **40.25** |
| hopper-medium-replay-v2 | 15.24 | **94.73** | 11.74 | **59.43** | 21.56 | **68.87** |
| walker2d-medium-replay-v2 | 4.67 | **81.72** | 6.89 | **45.64** | -0.25 | **53.52** |
| average | 7.16 | **74.51** | 6.73 | **47.09** | 6.57 | **54.21** |

Table 3: Comparing the performance of state-conditioned action discretization against unconditioned action discretization with CQL. The state-conditioned discretization scheme significantly outperforms the unconditioned one since unconditioned action discretization cannot compress the action space into few number of bins.

**Comparing discretization methods.** To understand the importance of the state-conditioned discretization method, we compare it against a naive discretization method where the VQ-VAE discretizes the actions without conditioning on the states and present the results in Table 3. We see that the state-conditioning allows is indeed highly important in compressing the action space into a small number of bins, resulting in much higher performance than a state-agnostic discretization scheme.

**Codebook size robustness.** One key design choice we make in this paper is the use of a VQ-VAE, one natural question is then our method's robustness against the codebook size in the VQ-VAE; since that can be crucial in determining the quality of the performed discretization through it. Towards this end, we empirically experiment with varying codebook sizes across all three algorithms. We present the results in Table 4, and we found that our method's performance is consistent across codebook sizes; which further resonates with the practical utility of adopting our method.

| Codebook Size | 16 | 32 | 64 | 128 |
|---|---|---|---|---|
| SAQ-CQL | 108.7 | 111.6 | 110.8 | 103.2 |
| SAQ-IQL | 106.9 | 104.8 | 104.2 | 94.42 |
| SAQ-BRAC | 106.5 | 108.3 | 105.7 | 107 |

Table 4: Comparing the performance of SAQ-IQL, SAQ-CQL, and SAQ-BRAC on hopper-expert-v2 while varying the codebook size. It can be observed that the discretized algorithms are invariant to codebook size changes.

**Controlling policy constraint levels.** As stated in Sec.4.1, one key premise of SAQ is that we can enforce policy constraint or value conservatism exactly; which associates with the practical performance of offline RL methods. To further verify this hypothesis empirically; we pick one task from the Gym locomotion suite and vary the weight coefficients for policy constraint or value conservatism to observe the resulting performance. We present our results in Fig. 5, we can see that small constraint enforcement leads to poor performance initially; then the performance ramps up when we increase the coefficients; finally converges with sufficient large coefficients. This observation confirms our conjecture in the paper.

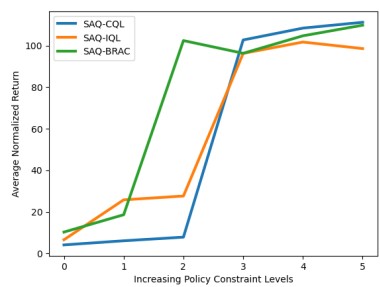

Figure 5: Increasing policy constraint levels on hopper-expert-v2 environment from Gym locomotion.

# C  Additional Experiment Results

| Task | BRAC | SAQ-BRAC | IQL | Aquadem-IQL | SAQ-IQL | CQL | Aquadem-CQL | SAQ-CQL | BC | Aquadem-BC | SAQ-BC |
|---|---|---|---|---|---|---|---|---|---|---|---|
| halfcheetah-expert-v2 | 67.99 | 64 | 94.78 | 92.73 | 90.3 | 43.9 | 92.66 | 92.65 | 91.53 | 92.69 | 108.4 |
| halfcheetah-medium-expert-v2 | 73.07 | 90.31 | 88.06 | 80.75 | 89.05 | 50.32 | 88.3 | 91.69 | 48.81 | 74.17 | 57.65 |
| halfcheetah-medium-replay-v2 | -3.3 | 35.28 | 44.24 | 35.12 | 36.2 | 47.07 | 40.5 | 40.25 | 35.16 | 35.62 | 3.76 |
| halfcheetah-medium-v2 | 47.08 | 43 | 47.3 | 42.39 | 42.52 | 48.56 | 44.5 | 43.89 | 42.32 | 41.93 | 47.02 |
| hopper-expert-v2 | 67.34 | 110 | 108.8 | 109 | 100.3 | 100.5 | 110.3 | 109.9 | 98.8 | 110.1 | 92.63 |
| hopper-medium-expert-v2 | 50.74 | 98.75 | 32.85 | 54.57 | 81.56 | 67.08 | 86.7 | 98.47 | 44.48 | 57.9 | 58.94 |
| hopper-medium-replay-v2 | 36.81 | 32.54 | 61.75 | 38.05 | 59.43 | 94.73 | 90.3 | 68.87 | 17.96 | 20.94 | 32.93 |
| hopper-medium-v2 | 57.06 | 54.16 | 54.3 | 49.2 | 50.53 | 70.32 | 58.5 | 38.25 | 50.31 | 52.98 | 42.47 |
| walker2d-expert-v2 | 108.4 | 107.5 | 110.12 | 108.5 | 107.6 | 109.2 | 108.2 | 107.3 | 108.3 | 108.3 | 104.5 |
| walker2d-medium-expert-v2 | 109 | 102.9 | 109.56 | 108 | 100.3 | 110.7 | 108.1 | 108.6 | 91.79 | 83.55 | 103.2 |
| walker2d-medium-replay-v2 | 73.72 | 0.4 | 68.71 | 32.66 | 45.64 | 81.72 | 80.8 | 53.52 | 12.98 | 27.86 | 15.27 |
| walker2d-medium-v2 | 81.94 | 64.8 | 81.25 | 68.06 | 68 | 83.11 | 82.1 | 74.77 | 70.18 | 27.86 | 62.52 |
| locomotion average | 64.16 | **66.98** | 75.14 | 71.01 | **76.67** | 75.6 | **82.58** | 77.35 | 59.39 | **61.16** | 60.77 |
| antmaze-medium-diverse-v2 | 26 | 47.6 | 76.67 | 40 | 68.33 | 72.75 | 22.67 | 75.47 | 0 | 0 | 0 |
| antmaze-medium-play-v2 | 48.67 | 56.93 | 78.67 | 46 | 74.33 | 67.04 | 30.67 | 68.67 | 0 | 1.3 | 0 |
| antmaze-large-diverse-v2 | 0.66 | 9.73 | 31.67 | 33.67 | 41 | 35.62 | 22.67 | 36 | 0 | 0 | 0 |
| antmaze-large-play-v2 | 0 | 3.73 | 34.33 | 14.33 | 40 | 45.18 | 25.33 | 47.33 | 0 | 0 | 0 |
| antmaze average | 18.84 | **29.5** | 55.34 | 33.5 | **55.92** | 55.15 | 25.34 | **56.87** | 0 | **0.33** | 0 |
| door-human-v0 | −1.01 | 35.42 | 1.79 | 2.15 | 9.26 | 0.84 | 6.09 | 2.12 | 3.29 | 3.24 | 9.28 |
| hammer-human-v0 | −1.42 | 20.52 | 1.41 | 1.56 | 1.57 | 0.27 | 2.64 | 0.6 | 0.8 | 2.09 | 1.38 |
| pen-human-v0 | 98.15 | 98.41 | 69.69 | 73.91 | 80.25 | 41.24 | 85.66 | 82.73 | 45.28 | 72.12 | 73.3 |
| relocate-human-v0 | −0.28 | 6 | 8.38 | 0.55 | 0.2 | −0.05 | 0.28 | 0.02 | 0.04 | 0.48 | 0.02 |
| adroit average | 23.86 | **40.09** | 20.32 | 19.54 | **22.82** | 10.58 | **23.67** | 21.37 | 12.35 | 19.48 | **21** |
| kitchen-mixed-v0 | 10.33 | 53.33 | 48.92 | 56.94 | 52.92 | 62 | 0 | 57.67 | 37.9 | 58.89 | 34 |
| kitchen-complete-v0 | 31.67 | 10 | 66 | 70.92 | 76.76 | 14 | 21.5 | 47.67 | 39.13 | 68 | 90.33 |
| kitchen-partial-v0 | 5.33 | 45 | 37.58 | 44.83 | 46.25 | 70 | 50 | 92.33 | 37.33 | 43.56 | 46.67 |
| kitchen average | 15.78 | **36.11** | 50.83 | 57.56 | **58.61** | 48.67 | 23.83 | **65.89** | 38.12 | 56.82 | **57** |

Table 5: Comparison of our method and Aquadem on various D4RL tasks. SAQ in general improves over its continuous counterpart, especially in the narrow dataset setting; also outperforms Aquadem in most tasks.

