# OpenReview forum: "Action-Quantized Offline Reinforcement Learning for Robotic Skill Learning"
_robot-learning.org/CoRL/2023/Conference — CoRL 2023 Poster_

### Official Review · Reviewer_g7YW · 2023-07-14

**Confidence:** 4
**Originality:** Good
**Technical Quality:** Fair
**Clarity Of Presentation:** Good
**Impact:** 3

**Recommendation:**

Weak Accept: I recommend accepting the paper, but will not argue for my recommendation if the majority of other reviewers have a different opinion.

**Review:**

**Strengths**

This paper presents an interesting approach to learn action discretization and presents useful empirical evidence showing that the discrete offline RL method variants outperform their continuous counterparts, sometimes significantly. Furthermore, this is shown across 3 different methods (CQL, IQL, and BRAC). As an added benefit, some of the modifications to the offline RL methods seem to simplify their implementation and allows for computing quantities exactly, due to the use of discrete actions.

**Weaknesses**

There are some issues with the experimental evaluation and the writing and clarity. Comments related to the experiments follow:

- The paper only compares the proposed method against the continuous action variants of the offline RL methods. However, there is limited analysis on whether the choice of VQ-VAE is important for the method to work. The VQ-VAE appears to be a crucial choice compared to other ways to discretize the actions, but that isn't really shown. The appendix does have some experiments showing that the state-conditioning for the VQ-VAE is important, and that the codebook size does not matter much. However, there are no comparisons to other discretization methods. Some other choices of discretization include the AQuaDem paper (https://arxiv.org/abs/2110.10149), which is highly related, and acknowledged in the related work section. It seems as though their discretization learning method could be applied to offline RL (even though they apply it to online RL). Other alternatives include using a categorical VAE with gumbel-softmax trick (https://arxiv.org/abs/1611.01144), and potentially learning a state-conditioned Gaussian Mixture Model (as in the robomimic paper, https://arxiv.org/pdf/2108.03298.pdf) and using the mode centers.

- Some of the robomimic results are questionable. Looking at Table 1 in the robomimic paper (https://arxiv.org/pdf/2108.03298.pdf), CQL obtains 92.7 and 38.0 respectively on the Lift (PH) and Can (PH) datasets, while the implementation in this paper achieves 64.2 and 19.6. The BC results in this paper are also significantly lower than the ones reported in the robomimic paper (e.g. Lift 100 vs. 59.5, Can 95.3 vs. 31.7, Square 78.7 vs. 19.3, Transport 17.3 vs. 0.3, and Tool Hang 29.3 vs. 1.87). Given that the implementation is open-source, it is hard to understand why these results are so much lower.

Some of the writing should be edited to improve clarity. Comments related to writing and clarify follow:

- The preliminaries section could be shortened - there are many details for CQL, BRAC, and IQL presented, and they are not all relevant to understand the core contribution of the method (using VQ-VAE to learn action discretization). The changes to each method are also highlighted in section 4.2 in a somewhat self-contained manner. It could be useful to make more explicit connections to the preliminaries section or just present all of the relevant details directly in section 4.2 and omit the sections in the preliminaries on these methods.

- Section 4.1 could be explained a little more clearly and precisely. For example, what is meant by single dimensional latent code? It most likely corresponds to K vectors in the codebook with D=1 (using the notation from the preliminaries section). The preliminaries section seems to have redundant information (for example equation 6 vs. equation 7 in this section). It would be better to make explicit connections to the preliminaries section to identify how the general VQ-VAE formulation was modified. This seems to be the core part of the method, so making it extremely clear is very important.

- The writing in Section 4.2 could also highlight the advantages of the use of discrete actions instead of continuous actions a little more. For example, it wasn't immediately clear that the CQL conservatism loss becomes an exact sum instead of an approximate sample-based integral. This would also help motivate the use of discrete actions over continuous actions more.

- Related to the point above, it might also help to provide some more intuition about the state-conditioned VQ-VAE. While the number of discrete latent actions is fixed, the interpretation of these actions depends on the state. In this sense, these latent codes could probably be thought of as modes of a conditional state-action distribution, similar to a Gaussian Mixture Model (GMM) or a categorical VAE. Given this choice, it seems a natural point of comparison could be to a GMM or categorical VAE (as mentioned in a comment above on adding more comparisons).

- It would be nice to see more useful analysis in section 5.1 and section 5.2 of the results in Table 1 and 2, with references to particular comparisons in the tables. For example, which improvements were the most significant and which were the least? Was this to be expected?

- More details on the specific implementation details should be provided. For example, what kinds of networks and hyperparameters were used? Did the BC baseline use a simple L2 loss, and how else did it differ from the one in the robomimic paper? How was the SAQ-BC model trained? Were the actions labeled into discrete codes and used to train a network to map states to discrete codes via a cross-entropy loss?

Some typos and potential mistakes:
- line 79 "actin"
- line 259 "re-visit once we have full results"

**Quality Of The Limitations Section:**

Limitations are addressed clearly

**Questions For Rebuttal:**

Please address the comments in the weaknesses section above.

**Robotics Focus:**

Highly relevant to robotics but no hardware experiments

**Summary Of Paper:**

This paper presents a method to learn a quantized action representation for use in continuous action offline reinforcement learning approaches. The method is to train a state-conditioned VQ-VAE to learn a set of discrete latent actions, which can be reconstructed by a state-conditioned decoder. Several offline RL methods are then adapted to use discrete actions instead of continuous actions. The action representation and discrete variants of the offline RL methods are evaluated against their continuous counterparts on two benchmarks - D4RL and robomimic, and shown to increase performance.

**Summary Of Recommendation:**

Currently, the issues with the experimental evaluation in the paper outweigh the positives. To summarize, the issues include not showing the value of the VQ-VAE compared to other discretization schemes and the poor results on the robomimic benchmark compared to the ones reported there.

Post-rebuttal: After the author rebuttal and the discussion, I have decided to raise my score to weak accept.

---

> ### Author Response · Authors · 2023-08-14
> **Deadline is tomorrow, friendly reminder**
>
> Dear reviewer g7YW,
>
> Thank you for your time to discuss with us.
>
> We sincerely appreciate if you can please let us know if we've addressed all of the concerns in your review, or if there is anything else we can add or clarify.
>
> Thank you again for taking the time to review our work. If you find our answers address your concerns, would you mind raising the score?
>
> Since the deadline is tomorrow, we'd really appreciate a reply.
>
> Best regards,
>
>  Authors

---

### Official Review · Reviewer_Aebd · 2023-07-19

**Confidence:** 4
**Originality:** Very Good
**Technical Quality:** Excellent
**Clarity Of Presentation:** Excellent
**Impact:** 4

**Recommendation:**

Strong Accept: I recommend accepting the paper and will argue for my recommendation even if other reviewers hold a different opinion.

**Review:**

## Strengths

1. SAQ is a simple and general method that can be applied on top of any base algorithm for offline RL. This makes its applicability potentially large and impactful.

2. The empirical results are clear and show substantial improvements in performance. These improvements seem to hold across most environments, but are especially strong on high dimensional tasks with more realistic demonstration data like robomimic, underscoring the potential importance for the CoRL audience.

## Weaknesses

1. The idea is not entirely original since Dadashi et al (cited in the paper as [29]) propose a very similar method for online RL. However, the paper provides this reference and builds on the idea to extend it into offline RL with a VQ-VAE.

2. Section 4.3 provides an explanation of why SAQ is helpful on top of CQL, but leaves a few lingering questions. (1) While it is clear that CQL makes an approximation with continuous actions, this explanation does not as cleanly extend to IQL or BC, but these methods also benefit from SAQ. Thus, it seems that this explanation may not be complete. (2) It is unclear how the exact CQL penalty is calculated. Is it just by taking many more samples so the estimate concentrates around the true value?

**Quality Of The Limitations Section:**

Limitations are addressed clearly

**Questions For Rebuttal:**

See weaknesses above

**Robotics Focus:**

Highly relevant to robotics but no hardware experiments

**Summary Of Paper:**

This paper presents state-conditioned action quantization (SAQ) a method for discretizing action spaces with a state-conditioned VQ-VAE. The method can be added on top of any standard offline RL algorithm to simplify the action space. Experiments are presented on a variety of base algorithms and tasks from D4RL and robomimic that show substantial improvements in performance by providing a simpler action space with a strong prior for actions in the dataset.

**Summary Of Recommendation:**

The paper presents a simple method that provides clear cut improvements over baseline methods and seems to be most effective on tasks that are most applicable to robotics. I think the paper is a good contribution and should be accepted.

---

### Official Review · Reviewer_Pinn · 2023-07-20

**Confidence:** 4
**Originality:** Good
**Technical Quality:** Fair
**Clarity Of Presentation:** Good
**Impact:** 2

**Recommendation:**

Weak Reject: I recommend rejecting the paper, but will not argue for my recommendation if the majority of other reviewers have a different opinion.

**Review:**

Overall, I liked the paper as it is interesting, clearly written, and shows significant improvements over the baselines. But, it is missing some analyses to understand the core issue in the continuous action spaces and why the discretization is performing much better.

# Strengths
- Clearly written, well-explained and the transformation of continuous to discrete versions of the algorithms is clear.
- The idea of using state-conditioned VQ-VAE to discretize the action space is quite interesting and it's nice to see it works well too in practice.
- The results often show a significant gain over the compared continuous action space counterparts. The results seem very strong.


# Weaknesses
My main concern is that the paper is **missing qualitative and quantitative investigation of the contributions of this paper**: 1. the use of VQ-VAE to discretize and 2. discrete v/s continuous action space in offline RL:

## VQ-VAE for discretization
Some implementation details and crucial analyses are missing:
- I wonder how difficult or unstable the training of VQ-VAE was, and is there any plan to open-source the code? What was the size of the codebook? Could the authors provide more training details, especially those pertaining to this paper's contributions.
- Is the discrete codebook sufficiently able to recover all the action reconstructions? What is the performance of the action reconstruction?
- What exactly is the role of the codebook and the VQ-VAE decoder? One trivial solution is that the VQ-VAE decoder learns to disregard the codebook and simply learns a BC policy $f(a | s, e_k) \sim f_{BC}(a | s)$. Since, the decoder essentially does BC and the other offline-RL methods now use this decoder implicitly, it is possible that their performance improves more on expert dataset, because of this inherent bias towards BC, by assuming that the data is good for imitation. I am not at all claiming that this happens, but understanding what is the learned discrete action space, codebook, and VQ-VAE encoder-decoder seems important to justifying where the performance gains come from.
- **Other discretization approaches**: The paper mentions several action discretization works [21-28] which either assume independence of action dimensions or model the action space autoregressively, and the claim is that naive discretization can result in exponential blowup or imprecise actions. While this makes sense intuitively, since the task is offline RL, the challenge of exploration is not there. This means that there is a chance that even with such a discrete action space, learning is feasible and at least easier than the continuous action space counterpart — for instance, if the discretized action space is quite large, then the sampling versions of the offline RL algorithms can still be applied. Nevertheless, since VQ-VAE as the discretization scheme is one of the core contributions, it is important to compare against alternate action discretization schemes to disentangle the roles of discretization scheme and the question of discrete v/s continuous offline RL optimization. I do appreciate the ablation of unconditioned VQ-VAE, but that only explores one style of discretization: reconstruction with VQ-VAE. It would be nice to see the effect of some other methods such as [23, 24, 25].

## Discrete v/s continuous action space for offline RL
It is great to see the improvement with the discretization, but the current experiments do not convincingly show that the benefit is actually because of replacing the approximate calculations with exact versions, or due to other artifacts of discrete action spaces, such as the ease of multi-modal modeling of data.
- Especially, the result of BC v/s SAQ-BC hints at the fact that there are major improvements even in the absence of any approximation made in the continuous action space.
	+ It's not even clear that multimodality is the core reason of improvement, because the SAQ-BC v/s BC gains are more significant in expert datasets (especially, halfcheetah and hopper) than medium datasets. It is unclear to me why SAQ-BC (92.63) would be so much better than BC (28.49) when the dataset is expert.
- In Fig 2, at around 1200 grad steps in continuous CQL achieves optimal performance despite the estimated R($\theta$) already diverging from the exact value. What explains this success despite the divergence? It would hint that even continuous action space offline RL can reach optimal performance despite the approximations over the constraints.
	+ Also, if instability only arises post optimality, then wouldn't early stopping be a simple solution to continuous CQL's issues?
	+ Can the authors provide the learning curves for all the results computed to observe the learning dynamics? Since the key claim is the poor optimization due to approximations in continuous action spaces, it is important to observe the performance over the course of training.


**Quality Of The Limitations Section:**

Additional details required

**Questions For Rebuttal:**

Several questions are raised in the review above. Additionally, I am curious why the authors mention that "the main benefit of SAQ is not in raw performance", despite the results showing large improvements over the traditional continuous action space way of solving the problem. Is there something that I missed about why the improvement in performance is not claimed more strongly? For example, I did not check the performance of these algorithms in other prior works. Do the authors suspect that the continuous action space results can improve with some implementation optimizations or better hyperparameter tuning, and the true difference in performance against the discrete action space results is smaller?

**Robotics Focus:**

Highly relevant to robotics but no hardware experiments

**Summary Of Paper:**

This proposes that offline RL methods like CQL, IQL, and BRAC employ a sampling step to optimize the constraints when the action space is continuous, making their performance poor. Thus, this paper discretizes the action space through state-conditioned reconstruction of action using a VQ-VAE (thus ensuring a small action space). The discretized versions of the above algorithms perform better than their continuous counterparts.

**Summary Of Recommendation:**

Overall, I liked the paper as it is interesting, clearly written, and shows significant improvements over the baselines. But, it is missing some analyses to understand the core issue in the continuous action spaces and why the discretization is performing much better. The current hypothesis that continuous action space offline RL suffers due to approximations in the estimation of the constraints is not empirically justified well. In this state, I cannot recommend acceptance despite the strong results the paper reports. I would be more than happy to reconsider my position if provided with more information such as implementation details, training curves, hyperparameter tuning approach, the author's answers, and analysis experiments.

---

> ### Comment · Reviewer_Pinn · 2023-08-12
> **Most concerns answered, but a couple remain.**
>
> I appreciate the authors providing evidence to alleviate several concerns. I am satisfied with VQ-VAE's training reliability, its use of the codebook, comparison to Aquadem.
>
> My two leftover concerns I have responded to the results comment:
> 1. While I don't expect all the learning curves, the authors can still provide some learning curves to demonstrate the exact issue that the continuous action space environments are having. If the only issue is instability post-optimal convergence, then I don't believe all this discretization is justified.
> 2. I don't understand the claim that early stopping "needs per-domain tuning thus making it not a very stable solution". If anything, it's the most basic tuning-free method that one would try on any supervised learning problem. The fact that early stopping was not performed to obtain the above results is surprising to me. Even this papers' results might increase with early stopping. By early stopping, I just mean selecting the best model checkpoint based on its performance on a validation dataset.
>
> I believe discretization has many benefits, and, in principle, I hope this paper's ideas work. But the reliability of experiments and the issue in continuous variants of these offline RL algorithms requires a proper investigation. I do not ask for complete results, but some learning curves to confirm that indeed discrete > continuous at optimality are necessary.

---

> ### Author Response · Authors · 2023-08-13
> **Addressing your concern**
>
> Dear reviewer Pinn,
>
> Thank you for your time for discussion with us.  To further address your concerns, we did early stopping ablations as you mentioned.
>
> While it's true that individual continuous offline RL algorithms could likely be improved by inventing early stopping rules based on, for example, the approximation error in the conservatism penalty, this is an open research direction. In general it's very hard in the offline setting because early stopping requires online evaluation. The closest prior work on this is Kumar et al. 2021 [1].
>
> We would certainly be happy to compare to these methods proposed in prior work, and we did run a comparison to the early stopping rule that we think you are proposing (to stop based on the difference between the estimated and actual conservatism penalty, i.e., the gap between blue and green in Fig 2). This improves performance a bit, as shown below, but doesn't close the gap to SAQ-CQL.
>
> Also, note that the early stopping rule you mentioned is an oracle-style stopping rule, the maximum performance would be undoubtfully better; that said the SAQ could also increase performance based on this oracle ckpt selection rule.
>
> Thus, we report following results of checkpoint selection comparison in robomimic: 1) CQL/SAQ-CQL checkpoint selection based on Kumar et al. 2021 2) CQL/SAQ-CQL checkpoint selection based on the rule you mentioned 3)original CQL/SAQ-CQL results in our paper 4) CQL results from robomimic paper
>
> In general, though, we would emphasize that be we do not believe that the benefit of SAQ in the long run will be to always improve over any continuous RL method: the community will inevitably invent better methods in the future. The benefit is that SAQ doesn't seem to require these approximations, early stopping rules, and other tricks to attain relatively reliable good performance. We think in the long run this is much more important than getting the absolute best results on every benchmark.
>
>
>  task        | original CQL results | CQL checkpoint selection based on highest return | CQL checkpoint selection based on Kumar et. al. (2021) | VQ-CQL (ours) | VQ-CQL checkpoint selection based on highest return | VQ-CQL checkpoint selection based on kumar et. al. (2021) |CQL results from robomimic paper |
> |-------------|-------------|--------------------------------------------------|----------------------------------------------------|--------------|----------------------------------------------------|-------------------------------------------------------|---------------|
> | lift        | 64.2        | 70                                               | 64.67                                              | 90.8         | 93.33                                              | 82                                                  | 92.7          |
> | can         | 19.6        | 27.33                                            | 19.6                                               | 71.2         | 74                                                 | 62.67                                               | 38            |
> | square      | 0           | 1                                                | 0                                                  | 44.27        | 51.33                                              | 51.33                                               | 5.3
> | tool hang      | 0           | 0                                           | 0                                                  | 3.87        | 8                                    | 1.33                                              | 0
> | transport      | 0           | 0                                           | 0                                                  | 3.47        | 4                                   | 2                                             | 0
> | Average      | 16.76           | 19.76                                           | 16.85                                                  | 42.72       | 46.13                                  | 39.87                                             | 27.2
>
> [1] Kumar et al. A Workflow for Offline Model-Free Robotic Reinforcement Learning  CoRL 2021
>
> Thank you for your time again, please let me know if I can further clarify.

---

### Official Review · Reviewer_L3mY · 2023-07-20

**Confidence:** 4
**Originality:** Good
**Technical Quality:** Very Good
**Clarity Of Presentation:** Very Good
**Impact:** 3

**Recommendation:**

Weak Accept: I recommend accepting the paper, but will not argue for my recommendation if the majority of other reviewers have a different opinion.

**Review:**

Strengths:
- The paper presents a novel solution to a challenging and practically useful problem
- The paper is clearly written and easy to understand.
- The method appears to provide substantial improvements over existing approaches.

Weaknesses:
- No D4RL experiments in the main paper.  The authors devote space to showing images from the D4RL benchmark (Figure 3) but not their actual results, which are instead in the appendix.
- The average improvement in each category in D4RL hides an interesting observation that deserves more attention: SAQ appears to slightly harm performance for many individual tasks, but drastically improves performance in  others.  The locomotion tasks especially exhibit this phenomenon, where for most tasks, performance is actually harmed by SAQ, even though the average scores are propped up by very large gains in a few specific tasks.  A more thorough analysis here would be extremely useful.  In these tasks where SAQ hurts performance, what is the failure mode?  Are the continuous actions in successful rollouts not covered by the discretization, or is something else happening?  This may require more space than could fit in the main paper, but would be really great supplemental analysis.
- A comparision against AQUADEM's log-sum-exp discretization technique seems important here.  Even though, AQUADEM is designed for demonstrations+RL and SAQ is designed for offline RL, they both share the technique of pretraining a state-dependent discretization, and so it would be extremely useful to know which discretization technique works better here.

Typos:
- "re-visit once we have full results" at the end of the 1st paragraph of 5.2

Post rebuttal: I am keeping weak-accept at this time.  The other reviewers found some inconsistencies in the experiments, which the authors attempted to correct.  It seems like the others disagree about whether or not these issues were properly addressed.  The issues of the other reviewers are somewhat concerning, but to me there are enough interesting components and ideas here  that I don't think it's worth moving my score down.  I generally buy Aebd's argument in response to g7YW: "I think that your robomimic concerns are orthogonal to the main point of the paper. In my mind, the point of the paper is to compare base algorithms (e.g. CQL, BC, IQL) with exactly the same algorithms plus SAQ. While they could tune hyperparameters for each domain like robomimic, it seems like a cleaner comparison between standard CQL and CQL+SAQ to use the simpler hyperparameter tuning strategy used in this paper. Moreover, if the hyperparameter tuning were that extensive, it would be the same for CQL+SAQ."

**Quality Of The Limitations Section:**

Limitations are addressed clearly

**Questions For Rebuttal:**

Some analysis of the issues mentioned above about SAQ harming performance by small amounts in many tasks, while improving performance by large amounts in a few tasks would be very helpful.

If you have time to run the ablation with AQUADEM's discretization method on a few tasks that would also be interesting to see.

**Robotics Focus:**

Highly relevant to robotics but no hardware experiments

**Summary Of Paper:**

The authors propose a new method for online reinforcement learning with continuous action spaces.  This method uses a VQ-VAE to learn a state-dependent discretization of the action space.  They then show that this discretization provides significant benefits to existing offline reinforcement learning techniques.

**Summary Of Recommendation:**

The paper demonstrates new practical tools for addressing important problems in the learning and robotics communities.  If I were starting a new project in this area in the near future, I would want to read this paper.

Post Rebuttal: see notes in main review above.

---

### Author Response · Authors · 2023-08-12
**Comparison to AquaDem**

| Task                                 | CQL   | VQ-CQL | Aquadem-CQL | IQL   | VQ-IQL | Aquadem-IQL | BC    | VQ-BC | Aquadem-BC |
|--------------------------------------|-------|--------|-------------|-------|--------|-------------|-------|-------|------------|
| door-human-v0                        | 0.84  | 2.12   | 6.09    | 1.79  | 9.26   | 2.15        | 3.29  | 9.28  | 3.24       |
| hammer-human-v0                      | 0.27  | 0.6    | 2.64        | 1.41  | 1.57   | 1.56        | 0.8   | 1.38  | 2.09       |
| pen-human-v0                         | 41.24 | 82.73  | 85.66       | 69.69 | 80.25  | 73.91       | 45.28 | 73.3  | 72.12      |
| relocate-human-v0                    | -0.05 | 0.02   | 0.28        | 8.38  | 0.2    | 0.55        | 0.04  | 0.02  | 0.48       |
| adroit average                       | 10.58 | 21.37  | 23.66       | 20.32 | 22.82  | 19.54       | 12.35 | 21    | 19.48      |
| kitchen-mixed-v0                     | 62    | 57.67  | 0           | 48.92 | 52.92  | 56.94       | 11.67 | 34    | 58.89      |
| kitchen-complete-v0                  | 14    | 47.67  | 21.5        | 66    | 76.67  | 70.92       | 16.67 | 90.33 | 68         |
| kitchen-partial-v0                   | 70    | 92.33  | 50          | 37.58 | 46.25  | 44.83       | 23.33 | 46.67 | 43.56      |
| kitchen average                      | 48.67 | 65.89  | 23.83       | 50.83 | 58.61  | 57.56       | 17.22 | 57    | 56.82      |
| walker2d-expert-v2                   | 109.2 | 107.3  | 108.2       | 110.12| 107.6  | 108.5       | 95.92 | 104.5 | 108.3      |
| walker2d-medium-expert-v2             | 110.7 | 108.6  | 108.1       | 109.56| 100.3  | 108         | 70.21 | 103.2 | 83.55      |
| walker2d-medium-replay-v2             | 81.72 | 53.52  | 80.8        | 68.71 | 45.64  | 32.66       | 11.36 | 15.27 | 27.86      |
| walker-2d-medium-v2                  | 83.11 | 74.77  | 82.1        | 81.25 | 68     | 68.06       | 60.18 | 62.52 | 27.86      |
| hopper-expert-v2                     | 100.5 | 109.9  | 110.3       | 108.8 | 100.3  | 109         | 28.49 | 92.63 | 110.1      |
| hopper-medium-expert-v2               | 67.08 | 98.47  | 86.7        | 32.85 | 81.56  | 54.57       | 38.09 | 58.94 | 57.9       |
| hopper-medium-replay-v2               | 94.73 | 68.87  | 90.3        | 61.75 | 59.43  | 38.05       | 14.1  | 32.93 | 20.94      |
| hopper-medium-v2                      | 70.32 | 38.25  | 58.5        | 54.3  | 50.53  | 49.2        | 47.92 | 42.47 | 52.98      |
| halfcheetah-expert-v2                 | 43.9  | 92.65  | 92.66        | 94.78 | 90.3   | 92.73       | 7.3   | 108.4 | 92.69      |
| halfcheetah-medium-expert-v2           | 50.32 | 91.69  | 88.3        | 88.06 | 89.05  | 80.75       | 36.43 | 57.65 | 74.17      |
| halfcheetah-medium-replay-v2           | 47.07 | 40.25  | 40.5        | 44.24 | 36.2   | 35.12       | 19.91 | 3.76  | 35.62      |
| halfcheetah-medium-v2                  | 48.56 | 43.89  | 44.5        | 47.3  | 42.52  | 42.39       | 36.39 | 47.02 | 41.93      |
| gym average                          | 75.6  | 77.35  | 82.58       | 75.14 | 76.67  | 71.01       | 38.86 | 60.77 | 61.16      |
| antmaze-medium-diverse-v2            | 72.75 | 75.47  | 22.67       | 76.67 | 68.33  | 40          | 0     | 0     | 0          |
| antmaze-medium-play-v2               | 67.04 | 68.67  | 30.67       | 78.67 | 74.33  | 46          | 0     | 0     | 1.3        |
| antmaze-large-diverse-v2             | 35.62 | 36     | 22.67       | 31.67 | 41     | 33.67       | 0     | 0     | 0          |
| antmaze-large-play-v2                | 45.18 | 47.33  | 25.33       | 34.33 | 40     | 14.33       | 0     | 0     | 0          |
| antmaze average                      | 55.15 | 56.87  | 25.34       | 55.34 | 55.92  | 33.5        | 0     | 0     | 0.33       |
| lift                                 | 64.2  | 90.8   | 89.33       | 58    | 94     | 90.67       | 59.47 | 90.13 | 85.33      |
| can                                  | 19.6  | 71.2   | 66.67       | 33.73 | 61.5   | 75.2        | 31.73 | 66.4  | 73.33      |
| square                               | 0     | 44.27  | 43.73       | 26.93 | 48     | 53.73       | 19.33 | 45.33 | 50         |
| tool hang                            | 0     | 3.87   | 2.53        | 2.67  | 28     | 7.87        | 1.87  | 3.47  | 9          |
| transport                            | 0     | 3.47   | 2.36        | 0     | 2.67   | 3.6         | 0.27  | 3.2   | 4          |
| robomimic average                    | 16.76 | 42.72  | 40.92       | 24.27 | 46.83  | 46.21       | 22.53 | 41.71 | 44.33      |
| Average over different task suites   | 50    | 59.01  | 56.23       | 52.79 | 57.01  | 51.25       | 24.29 | 42.6  | 43.04      |
| Combined total (sum of scores on all tasks) | 1399.9 | 1652.38 | 955.92  | 1478.16 | 1596.38 | 1434.96 | 680.05 | 1192.8 | 1205.24 |

---

> ### Comment · Reviewer_Pinn · 2023-08-12
> **Many scores of baselines are much lower than the reported papers**
>
> At a glance, I was happy to see the great results of this paper. But after careful investigation and Reviewer g7YW's note on robomimic result inconsistencies, I now notice that many numbers for CQL, BC, IQL are lower than the numbers reported in the prior papers themselves. Since this is simply learning from offline data, these discrepancies are not justified as much as they might have been in online RL. A simple fix would be to use the max{authors' implementation, paper's reported numbers} for the baselines and then perform this comparison.
>
> For instance, BC and CQL numbers in [IQL](https://arxiv.org/pdf/2110.06169.pdf) paper are way higher than this paper reports. I tried to recompute the numbers after corrections, but there were way too many differences to count. This is a serious issue and the authors' jobs to ensure the correct numbers are reported where the baselines are well optimized and/or results are reported from the prior works. I would recommend authors survey more recent papers that report numbers on the same benchmarks as these papers and update this table properly. At least CQL and IQL papers are there where the numbers should lower bound this papers' numbers.
>
> While one might claim that for fair comparison, everything is run on the new code, there is always the unintentional possibility that the optimizations made benefit the discrete action-space learning dynamics (which is why aquadem also performs well) and not the continuous action-space dynamics, which are arguably expected to be quite different. Therefore, the best reporting style would be max{authors' implementation, paper's reported numbers}.
>
>
> Looking at how the Figure 2 experiment was conducted, I suspect there might be simple implementation fixes that all the implementations might be missing, but the baselines are getting adversely affected, such as early stopping. I don't understand the authors' claim that early stopping "needs per-domain tuning thus making it not a very stable solution." And without the learning curves (which understandably could not be generated given the limited time), it's hard to understand what happened in the learning dynamics of the continuous-action space offline RL methods that made them suboptimal to the discrete action variants.

---

> > ### Author Response · Authors · 2023-08-13
> > **Scores are NOT much lower than reported papers,  clarification 1/2**
> >
> > Thank you for your comments, we apologize the continuous BC results in some environments may not exactly align with IQL paper as you pointed out, we didn't do extensive tuning on BC during the submission; we'll make sure to update those results in the final version of the paper. However we want to point out that our **continuous CQL/IQL results are actually ALIGNED with the IQL paper (Kostrikov. el al. 2021) [1]**.  For your reference, we report results from our paper, results from IQL paper, and SAQ results below.
> >
> > | Task                         | CQL    | CQL (Kostrikov et al.) | SAQ-CQL | IQL    | IQL (Kostrikov et al.) | SAQ-IQL | BC    | BC (Kostrikov et al.) | SAQ-BC |
> > | ---------------------------- | ------ | ---------------------- | ------- | ------ | ---------------------- | ------- | ----- | --------------------- | ------ |
> > | door-human-v0                | 0.84   | 9.9                    | 2.12    | 1.79   | 4.3                    | 9.26    | 3.29  | 2                     | 9.28   |
> > | hammer-human-v0              | 0.27   | 4.4                    | 0.6     | 1.41   | 1.4                    | 1.57    | 0.8   | 1.2                   | 1.38   |
> > | pen-human-v0                 | 41.24  | 37.5                   | 82.73   | 69.69  | 71.5                   | 80.25   | 45.28 | 63.9                  | 73.3   |
> > | relocate-human-v0            | \-0.05 | 0.2                    | 0.02    | 8.38   | 0.1                    | 0.2     | 0.04  | 0.1                   | 0.02   |
> > | adroit average               | 10.58  | 13                     | 21.37   | 20.32  | 19.33                  | 22.82   | 12.35 | 16.8                  | 21     |
> > | kitchen-mixed-v0             | 62     | 51                     | 57.67   | 48.92  | 51                     | 52.92   | 11.67 | 51.5                  | 34     |
> > | kitchen-complete-v0          | 14     | 43.8                   | 47.67   | 66     | 62.5                   | 76.67   | 16.67 | 65                    | 90.33  |
> > | kitchen-partial-v0           | 70     | 49.8                   | 92.33   | 37.58  | 46.3                   | 46.25   | 23.33 | 38                    | 46.67  |
> > | kitchen average              | 48.67  | 48.2                   | 65.89   | 50.83  | 53.27                  | 58.61   | 17.22 | 51.5                  | 57     |
> > | walker2d-medium-expert-v2    | 110.7  | 108.8                  | 108.6   | 109.56 | 109.6                  | 100.3   | 70.21 | 107.5                 | 103.2  |
> > | walker2d-medium-replay-v2    | 81.72  | 77.2                   | 53.52   | 68.71  | 73.9                   | 45.64   | 11.36 | 26                    | 15.27  |
> > | walker2d-medium-v2           | 83.11  | 72.5                   | 74.77   | 81.25  | 78.3                   | 68      | 60.18 | 75.3                  | 62.52  |
> > | hopper-medium-expert-v2      | 67.08  | 105.4                  | 98.47   | 32.85  | 91.5                   | 81.56   | 38.09 | 52.5                  | 58.94  |
> > | hopper-medium-replay-v2      | 94.73  | 95                     | 68.87   | 61.75  | 94.7                   | 59.43   | 14.1  | 18.1                  | 32.93  |
> > | hopper-medium-v2             | 70.32  | 58.5                   | 38.25   | 54.3   | 66.3                   | 50.53   | 47.92 | 52.9                  | 42.47  |
> > | halfcheetah-medium-expert-v2 | 50.32  | 91.6                   | 91.69   | 88.06  | 86.7                   | 89.05   | 36.43 | 55.2                  | 57.65  |
> > | halfcheetah-medium-replay-v2 | 47.07  | 45.5                   | 40.25   | 44.24  | 44.2                   | 36.2    | 19.91 | 36.6                  | 3.76   |
> > | halfcheetah-medium-v2        | 48.56  | 44                     | 43.89   | 47.3   | 47.4                   | 42.52   | 36.39 | 42.6                  | 47.02  |
> > | gym average                  | 72.62  | 77.61                  | 68.7    | 65.34  | 76.96                  | 67.58   | 37.18 | 51.86                 | 47.08  |
> > | antmaze-medium-diverse-v2    | 72.75  | 53.7                   | 75.47   | 76.67  | 70                     | 68.33   | 0     | 0                     | 0      |
> > | antmaze-medium-play-v2       | 67.04  | 61.2                   | 68.67   | 78.67  | 71.2                   | 74.33   | 0     | 0                     | 0      |
> > | antmaze-large-diverse-v2     | 35.62  | 14.9                   | 36      | 31.67  | 47.5                   | 41      | 0     | 0                     | 0      |
> > | antmaze-large-play-v2        | 45.18  | 15.8                   | 47.33   | 34.33  | 39.6                   | 40      | 0     | 0                     | 0      |
> > | antmaze average              | 55.15  | 36.4                   | 56.87   | 55.34  | 57.08                  | 55.92   | 0     | 0                     | 0      |
> > | average of all tasks         | 53.13  | 52.04                  | 56.45   | 52.16  | 57.9                   | 53.2    | 21.78 | 34.42                 | 33.94  |
> >
> > [1]Offline Reinforcement Learning with Implicit Q-Learning

---

> > > ### Author Response · Authors · 2023-08-13
> > > **Clarification 2/2**
> > >
> > > It's worth noting that we performed experiments on an additional environment *-expert-v2*, which is different from *-medium-expert-v2*, this might cause misunderstandings; because Kostrikov et al. 2021 didn't include experiments on this environment.  For an apple to apple comparison, we also exclude that in the result table above, but we report the result for that environment below.
> > >
> > >
> > > | Task                  | CQL   | SAQ-CQL | IQL    | SAQ-IQL | BC    | SAQ-BC |
> > > | --------------------- | ----- | ------- | ------ | ------- | ----- | ------ |
> > > | walker2d-expert-v2    | 109.2 | 107.3   | 110.12 | 107.6   | 95.92 | 104.5  |
> > > | hopper-expert-v2      | 100.5 | 109.9   | 108.8  | 100.3   | 28.49 | 92.63  |
> > > | halfcheetah-expert-v2 | 43.9  | 92.65   | 94.78  | 90.3    | 7.3   | 108.4  |
> > > | average               | 84.53 | 103.28  | 104.57 | 99.4    | 43.9  | 101.84 |
> > >
> > >
> > > For robomimic results, as we stated in the paper, we didn't not perform extensive hyperparameter tuning nor using complex policy class,  so that's reason the BC number there is lower. For consistency, we also report robomimc paper results, our results, SAQ results below.
> > >
> > > | task      | BC   | robomimic BC | SAQ-BC  | CQL   | robomimic CQL | SAQ-CQL | IQL   | SAQ-IQL |
> > > |-----------|-------|--------------|--------|-------|---------------|-------|-------|-------|
> > > | lift      | 59.47 | 100          | 90.13  | 64.2  | 92.7          | 90.8  | 58    | 94    |
> > > | can       | 31.73 | 95.3         | 66.4   | 19.6  | 38            | 71.2  | 33.73 | 61.5  |
> > > | square    | 19.33 | 78.7         | 45.33  | 0     | 5.3           | 44.27 | 26.93 | 48    |
> > > | tool hang | 1.87  | 17.3         | 3.47   | 0     | 0             | 3.87  | 2.67  | 28    |
> > > | transport | 0.27  | 29.3         | 3.2    | 0     | 0             | 3.47  | 0     | 2.67  |
> > > | average   | 22.53 | 64.12        | 41.71  | 16.76 | 27.2          | 42.72 | 24.27 | 46.83 |
> > >
> > >
> > > We can see that even CQL number in robomimic paper outperformed our continuous CQL number, SAQ-CQL still outperformed the the best continuous CQL results from robomimic paper.
> > >
> > > In general though, we would emphasize that be we do not believe that the benefit of SAQ in the long run will be to always improve over any continuous RL method: the community will inevitably invent better methods in the future. The benefit is that SAQ doesn't seem to require these approximations, early stopping rules, and other tricks to attain relatively reliable good performance. We think in the long run this is much more important than getting the absolute best results on every benchmark.

---

> > > > ### Author Response · Authors · 2023-08-13
> > > > **Early stopping**
> > > >
> > > > While it's true that individual continuous offline RL algorithms could likely be improved by inventing early stopping rules based on, for example, the approximation error in the conservatism penalty, this is an open research direction. In general it's very hard in the offline setting because early stopping requires online evaluation. The closest prior work on this is Kumar et al. 2021 [1].
> > > >
> > > > We would certainly be happy to compare to these methods proposed in prior work, and we did run a comparison to the early stopping rule that we think you are proposing (to stop based on the difference between the estimated and actual conservatism penalty, i.e., the gap between blue and green in Fig 2). This improves performance a bit, as shown below, but doesn't close the gap to SAQ-CQL.
> > > >
> > > > Also, note that the early stopping rule you mentioned is an oracle-style stopping rule,  the maximum performance would be undoubtfully better; that said the SAQ could also increase performance based on this oracle ckpt selection rule.
> > > >
> > > > Thus, we report following results of checkpoint selection comparison in robomimic: 1) CQL/SAQ-CQL checkpoint selection based on Kumar et al. 2021 2) CQL/SAQ-CQL checkpoint selection based on the rule you mentioned 3)original CQL/SAQ-CQL results in our paper 4) CQL results from robomimic paper
> > > >
> > > >
> > > >
> > > >  task        | original CQL results | CQL checkpoint selection based on highest return | CQL checkpoint selection based on Kumar et. al. (2021) | VQ-CQL (ours) | VQ-CQL checkpoint selection based on highest return | VQ-CQL checkpoint selection based on kumar et. al. (2021) |CQL results from robomimic paper |
> > > > |-------------|-------------|--------------------------------------------------|----------------------------------------------------|--------------|----------------------------------------------------|-------------------------------------------------------|---------------|
> > > > | lift        | 64.2        | 70                                               | 64.67                                              | 90.8         | 93.33                                              | 82                                                  | 92.7          |
> > > > | can         | 19.6        | 27.33                                            | 19.6                                               | 71.2         | 74                                                 | 62.67                                               | 38            |
> > > > | square      | 0           | 1                                                | 0                                                  | 44.27        | 51.33                                              | 51.33                                               | 5.3
> > > > | tool hang      | 0           | 0                                           | 0                                                  | 3.87        | 8                                    | 1.33                                              | 0
> > > > | transport      | 0           | 0                                           | 0                                                  | 3.47        | 4                                   | 2                                             | 0
> > > > | Average      | 16.76           | 19.76                                           | 16.85                                                  | 42.72       | 46.13                                  | 39.87                                             | 27.2
> > > >
> > > > [1] Kumar et al. A Workflow for Offline Model-Free Robotic Reinforcement Learning  CoRL 2021

---

> ### Comment · Reviewer_Pinn · 2023-08-14
> **Summarized interpretation of the many changes after rebuttal**
>
> I sincerely appreciate the updates from the authors during the rebuttal period. Since there have been so many changes that were hard to keep track of, let me try to summarize my understanding here now:
> 1. SAQ-CQL >= CQL consistently, including on Robomimic experiments.
> (I agree with the authors' comment that early stopping is non-trivial and appreciate the new results too)
>
> 2. SAQ-BC < BC.
> - Since this paper's BC runs are very suboptimal to prior BC results on gym and Robomimic tasks, I will only check the comparison to prior BC results as this paper's BC implementation is not reliable.
> - Specially noting that SAQ-BC << BC on Robomimic (as g7YW pointed out)
>
> 3. SAQ-IQL <= IQL
> - Again, on the benchmarks common with IQL paper itself (57.9), SAQ-IQL does not improve (53.2).
>
> Therefore, many of the paper's original claims now need to be re-evaluated. I agree that there are significant improvements over CQL, but NOT over IQL and BC which the paper originally claims. That sums to 2/3rd of the paper's claims that need to be updated. It would not be right to accept the paper in its current form with the promise that 2/3rd of the paper will be revised without looking at what the new version would be. And that would require a new review totally.
>
> Keeping the fact that so much changed during rebuttal aside, even a paper that improves just over CQL with SAQ-CQL is a good paper. However, there is not enough empirical evidence to conclusively understand why CQL fails and that discretization is the key solution for that.
>
> Therefore, I would keep my recommendation of Weak Reject and would advise the authors to make the following changes in their revision:
> - Re-write the paper with claims centered around CQL that can be experimentally verified. BC and IQL results are good to show, but what the current paper claims are not justified by experimental performance.
> - Figure out the source of discrepancy in the BC and IQL implementations as compared to prior works. If numbers on past benchmarks match, then the other results of BC and IQL based on this paper's implementations will also be reliable.
> - More analyses like Figure 2 are necessary to really understand why CQL fails, and it is indeed the improved penalty computation over discrete action space that makes the optimization stable. A good start would be looking at the training curves.

---

> ### Author Response · Authors · 2023-08-14
> **Further clarification on claims**
>
> Dear reviewer,
>
> We really appreciate your time helping us to improve the paper.
>
> We agree with you that some of our gym BC results are not optimal, we will update the paper to reflect that.  For robomimic BC results,  as we also pointed out in our paper in line 254
>
> *It’s worthwhile to mention that our continuous BC results don’t exactly match the original paper [6], as the authors point out, they adopt a Gaussian Mixture Model(GMM) for the policy class and extensively optimize parameters then perform checkpoint selection; where we directly train uni-modal BC policies and perform reasonable parameter sweep. However we do observe that SAQ-BC is able to improve over BC substantially, this is because SAQ-BC is multi-modal, resonating with the continuous GMM BC results in the original paper.*
>
> We'll add the robomimic BC results to the paper's final version.
>
> **Regarding our claims**
>
> We respectfully disagree with what you mentioned "2/3 of the paper's claims need to be changed",  because it's not true.
>
> First, the paper is about improving offline RL algorithms, **NOT** about BC. BC is just a baseline for reference, has nothing to do with our claims.
>
> Second, we presented three offline algorithms, **BRAC, CQL, IQL**; it improves BRAC/CQL across the board (which you agreed).
> For IQL, on Robomimic, our results are much better (+92.95%); for D4RL benchmark tasks, as you requested, we pasted results from Kostrikov 2021, our continuous IQL is 52.16, theirs is 57.9 which is similar difference given codebase/implementation deltas, this small gap means the method is working;  and our discrete results is 53.2 > 52.16; however the improvement is much more pronounced on robomimic. The claims stay largely the same, we are not sure about the many changes you mentioned.  We also provided additional experiments to verify the effectiveness of the proposed method as you suggested (AquaDem, early stopping); we'd be happy to include this in the final version of our paper.
>
> Please let us know if we can further answer your concerns, thank you again for your time engaging with us.
>
> Best,
>
> Authors

---

### Decision · Program_Chairs · 2023-08-30

**Decision:**

Accept (Poster)

**Comment:**

This paper proposes action-discretization based on VQ-VAE to improve offline RL algorithms. The proposed method was combined with offline RL methods such as IQL and CQL, and the experimental results show that the proposed action discretization improves the performance of offline RL methods.

While the reviewers pointed out several unclear points, the authors addressed them in the rebuttal. Based on the overall assessment, AE recommends the acceptance of the paper.
However, Reviewer Pinn pointed out that the authors do not provide the learning curves of the proposed method. Please consider adding some learning curves and further analyzing what happens in the proposed method.